# Optimized LiFePO₄-Based Cathode Production for Lithium-Ion Batteries through Laser- and Convection-Based Hybrid Drying Process

Sebastian Wolf [1,*], Niklas Schwenzer [1], Tim Tratz [1], Vinzenz Göken [2], Markus Börner [2], Daniel Neb [1], Heiner Heimes [1], Martin Winter [2] and Achim Kampker [1]

[1] Production Engineering of E-Mobility Components (PEM), RWTH Aachen University, Bohr 12, 52072 Aachen, Germany
[2] Münster Electrochemical Energy Technology (MEET), Westfälische Wilhelms-Universität Münster, Corrensstraße 46, 48149 Münster, Germany
* Correspondence: s.wolf@pem.rwth-aachen.de

**Abstract:** The drying of electrodes for lithium-ion batteries is one of the most energy- and cost-intensive process steps in battery production. Laser-based drying processes have emerged as promising candidates for electrode manufacturing due to their direct energy input, spatial homogeneity within the laser spot, and rapid controllability. However, it is unclear to what extent electrode and cell quality are affected by higher heating and drying rates. Hybrid systems as a combination of laser- and convection-based drying were investigated in an experimental study with water-processed LFP cathodes. The manufactured electrodes were compared with purely laser-dried and purely convection-dried samples in terms of drying times and quality characteristics. The electrodes were characterized with regard to physical properties like adhesion and electronic conductivity, as well as electrochemical performance using the rate capability. Regarding adhesion and electronic conductivity, the LFP-based cathodes dried in the hybrid-drying process by laser and convection showed similar quality characteristics compared to conventionally dried cathodes, while, at the same time, significantly reducing the overall drying time. In terms of electrochemical performance, measured by the rate capability, no significant differences were found between the drying technologies used. These findings demonstrate the great potential of laser- and convection-based hybrid drying of LFP cathodes to enhance the electrode-drying process in terms of energy efficiency and operational costs.

**Keywords:** laser drying; convection drying; hybrid drying; LFP cathodes; lithium-ion battery production; electrode manufacturing; drying times





## 1. Introduction

Given the current economic and environmental challenges of lithium-ion battery production for electric vehicles, convective electrode drying plays a crucial role. Convection drying is the most energy-intensive process step in battery cell production, accounting for 27 to 47% of total energy consumption [1,2]. Moreover, convection drying is the most cost-intensive process step, accounting for 20–25% of the capital expenditures (CapEx) of battery production [3]. Furthermore, high operational expenditures (OpEx) result from the substantial energy demand and the high CapEx are attributed to dryer lengths of up to 100 m [4]. The already high space requirements further increase with higher production speeds and, thus, limit the scalability of electrode production [5,6].

Laser-based drying is a promising process alternative to convection-based drying that has been developed and investigated in recent years [6]. The direct and high energy input into the material through the absorption of laser radiation holds the potential for reduced energy consumption and space requirements by accelerated drying rates [7–9]. Substituting a share of convection drying with laser drying could consequently lead to

reduced greenhouse gas emissions, decreased CapEx and OpEx, and the enhanced scalability of electrode production. Moreover, laser drying allows for improved controllability of the drying process due to the low switch-on times reducing machine ramp-up compared to convection drying [10]. Inhomogeneous drying results often occurring in convection drying can potentially be mitigated by the homogeneous power density over the entire laser spot. Furthermore, the use of electricity to power laser modules avoids direct reliance on fossil fuels like natural gas.

However, a challenge for the large-scale deployment of laser drying is that the reduced space requirements are generally achieved through increased drying rates that can potentially cause detrimental degradation effects within the coating layer. Irrespective of the drying technology, high drying rates have repeatedly been shown to amplify the migration of binders and carbon black towards the electrode surface, causing the poor mechanical and electrochemical properties of dried electrodes [11–13]. This limits the applicability of laser drying on an industrial scale.

JAISER et al. have investigated binder migration within electrodes during the drying process [14]. Their findings suggest that higher drying rates are feasible in the initial stages of the drying process without compromising the mechanical and electrochemical properties of the electrode. This can be attributed to the fact that binder and carbon black migration is primarily caused by the capillary transport of the solvent towards the electrode surface, which predominantly occurs in later drying phases [11]. Therefore, a hybrid concept with an initial laser-based drying process at a high drying rate and subsequent convection-based drying at a low drying rate were investigated within this study. This aims to combine the potential advantages of laser drying in terms of reduced costs due to energy savings with convection drying while achieving a high electrode quality at increased production speeds.

## 2. Materials and Methods

### 2.1. Materials and Mixing

A cathode electrode paste was prepared and used throughout all experiments. The slurry consists of lithium iron phosphate (LFP, IBUvolt LFP400, IBU-tec Advanced Materials AG, Weimar, Germany) as active material and a binder system comprising the thickener carboxymethyl cellulose (CMC, MAC500LC, Nippon Paper Industries Co., Tokio, Japan), and the adhesion promoting latex binder styrene-butadiene rubber with 40% solid content in pre-dissolved solution (SBR, BM-451B, Zeon Europe GmbH, Dusseldorf, Germany). Furthermore, carbon black (C-Nergy Super C45, Imerys S.A., Paris, France) was added as conductive additive, and deionized water was used as solvent. The composition of the slurry is shown in Table 1.

**Table 1.** Composition of the LFP-based cathode slurry.

|  | LFP | SBR | CMC | Carbon Black | Solvent |
|---|---|---|---|---|---|
| $w_{solid}$ (%) [1] | 86.0 | 2.7 | 2.0 | 9.3 | 0.0 |
| $w_{total}$ (%) [2] | 38.0 | 3.0 | 0.9 | 4.1 | 54.0 |

[1] Mass percentage of solid components. [2] Total mass percentage of solid and liquid components.

In the mixing process, all components are, first, weighed with a precision scale inside a separated atmosphere to maintain a tolerance of $\pm 0.1$ g. The mixing tool and container of the deployed intensive mixer (EL 1, Maschinenfabrik Gustav Eirich GmbH & Co. KG, Hardheim, Germany) work in opposite directions of rotation. First, the dry components (CMC, conductive carbon black, and active material) are mixed at 350 rpm (1.5 m/s) for 15 min, and then at 500 rpm (2.1 m/s) for 5 min to achieve a uniform mixture of the components. In a separate container, the SBR and distilled water are mixed at 300 rpm (1.3 m/s) for 5 min to create a homogeneous liquid solution. A quarter of the dry mixture is then added to the solution and mixed at 1050 rpm (4.4 m/s) for 5 min. This is repeated three times. Intensive mixing at 2000 rpm (8.4 m/s) for 60 min completes the mixing process. The

container's side walls and mixing tool are scraped off between each mixing step to prevent agglomerate formation. To prevent sedimentation and phase separation in the setup time for the coating and drying equipment, the slurry is then mixed at 300 rpm (1.3 m/s) until the coating process begins.

### 2.2. Experimental Setup and Process Parameters

After mixing, the slurry is fed to the tank of a feed pump of a coating and drying system (2000K1, Robert Bürkle GmbH, Freudenstadt, Germany). The system can achieve a maximum web speed of 4 m/min. A slot die (SD250 4109-X, FMP Technology GmbH, Erlangen, Germany) is used for one-sided coating on 15 µm-thick aluminum foil (Speira GmbH, Grevenbroich, Germany). The slot die is operated at a coating width of 160 mm and a slot-die-to-substrate distance of 200 µm. Subsequently, the coated film is conveyed into the drying areas of the system, where either laser drying, convection drying, or both drying systems are active, depending on the process control. The coating and drying system is illustrated in Figure 1.

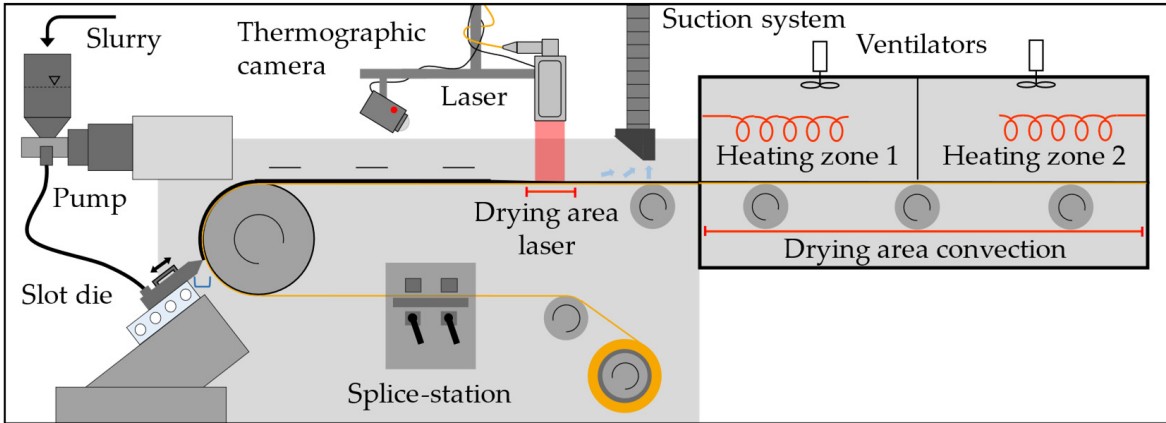

**Figure 1.** Schematic structure of the coating and drying process for hybrid drying [7].

The laser system features a diode laser (LDM 8000 W, Laserline GmbH, Mülheim-Kärlich, Germany) emitting radiation at two wavelengths of approximately 950 nm and 1000 nm. The zoom optics (OTZ-5 VR, Laserline GmbH) focus the radiation onto the 160 mm (width) × 170 mm (length) processing plane with a top-hat intensity distribution. A thermographic camera (Xi 400, Optris GmbH, Berlin, Germany) is used for process control and thermographic analysis of the laser-drying process. For convection drying, a 2.1 m dryer comprising two heating zones is used. Each heating zone is adjustable within a temperature range of 25 to 160 °C. To prevent possible influences on the electrode quality of the subsequent calendering process step, all tested samples are taken directly after the drying step in uncalendered condition for the quality tests.

The process parameters employed in the coating and drying process are described in detail in Table 2. To ensure a consistent wet film thickness (WFT) of 160 µm across all process parameters, the pump speed is individually adjusted in accordance with the web speed. It is crucial to note that, for convective drying in the given experimental setup, the maximum attainable web speed is 2 m/min. When exceeding this threshold, the coating fails to dry sufficiently within the given dryer length, heating zone temperatures, and WFT. In the case of stand-alone laser drying, the maximum attainable web speed is 1 m/min. To receive sufficiently dry results at higher web speeds, while keeping other factors constant (ceteris paribus), it becomes necessary to increase the laser power and, thus, enhance the drying rate. However, this approach results in significant drawbacks in terms of surface quality, as clearly visible cracks form on the coating. Consequently, the drying experiments for both convection and laser drying are conducted at a web speed reduced by 0.2 m/min from the maximum attainable web speed, to create an objective benchmark.

**Table 2.** Process parameter sets for coating and drying.

| Process Parameters | Convection | Laser | Hybrid | | |
|---|---|---|---|---|---|
| | I | II | III | IV | V |
| Web speed (m/min) | 1.8 | 0.8 | 2.8 | 4.0 | 2.8 |
| Pump speed (rpm) | 290 | 140 | 440 | 640 | 440 |
| Wet film thickness (µm) | 160 | 160 | 160 | 160 | 160 |
| Drying time laser/oven (s) | 0/70.0 | 12.8/0 | 3.6/45.0 | 2.6/31.5 | 3.6/22.5 |
| Oven heating zone 1 (°C) | 160 | <25 | 160 | 160 | 160 |
| Oven heating zone 2 (°C) | 140 | <25 | 140 | 140 | <25 |
| Laser power (W) | - | 456 | 736 | 1082 | 736 |
| Laser intensity (W/cm$^2$) | - | 1.68 | 2.71 | 3.98 | 2.71 |
| Laser energy input (J/cm$^2$) | - | 21.5 | 9.76 | 14.3 | 9.76 |

In the context of hybrid drying, the question at hand is the optimal configuration of laser power and oven temperature. For this, the thermographic analysis of stand-alone laser drying in a roll-to-roll process without convection drying shown in former publications is considered [15]. Therein, thermal imaging cameras were used to visualize four characteristic drying zones during the laser-based drying process (Cf. Figure 2a). Initially, the wet film reaches the laser spot at ambient temperature, and is then heated to about 90 °C in the preheating zone. Subsequently, a state of quasi-equilibrium is established between the energy introduced by the laser and the energy transferred to evaporation, which results in a constant drying rate and coating temperature of approximately 90 to 100 °C. The quasi-equilibrium state persists as long as the coating surface remains sufficiently wetted by capillary transport and until film shrinkage is completed [11,16]. The equilibrium state accounts for the majority of the evaporation process [17,18], henceforth referred to as the evaporation zone in this study. When the capillary transport no longer sufficiently wets the coating surface, pore dry-out begins. At this stage, the isolated solvent residues must overcome an additional transport resistance through the gas phase of the empty pores [19]. This leads to reduced mass transport, a falling drying rate, and an increasing coating temperature. This state is referred to as the overheating zone. Finally, the coating exits the laser spot and begins to slowly cool to ambient temperature in the cooling zone.

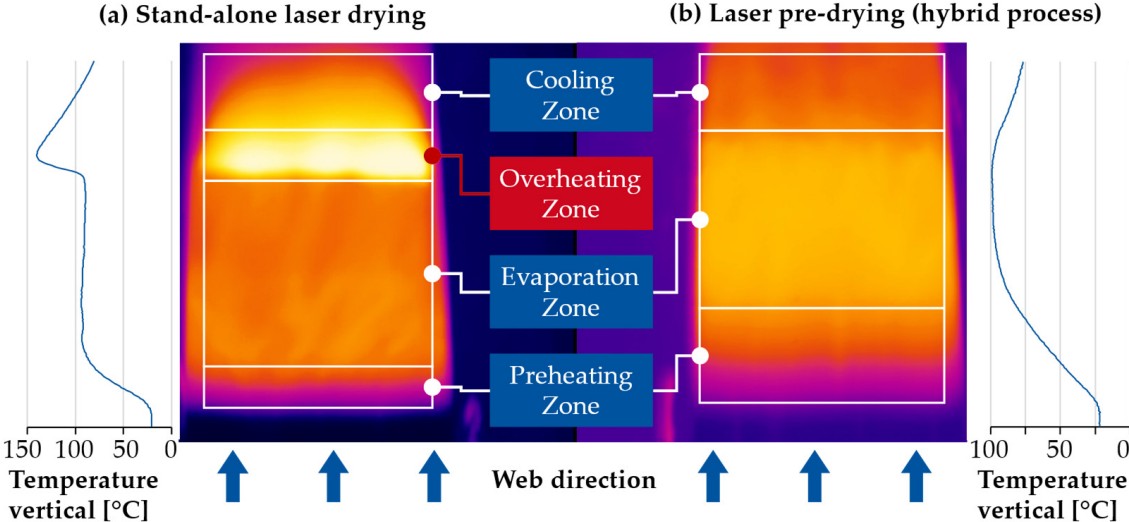

**Figure 2.** Characteristic temperature zones according to the thermographic monitoring of (**a**) stand-alone laser drying and (**b**) laser pre-drying in a hybrid-drying process.

In this study, the images of the integrated thermographic camera are used for process control during hybrid drying. The laser power is adjusted in accordance with the web speed

to prevent the slurry from exceeding the evaporation zone and forming an overheating zone (Cf. Figure 2b). In this way, the high drying rates of laser drying are only applied before pore emptying begins. The objective is to employ laser drying solely during film shrinkage before significant capillary transport effects occur. Consequently, laser intensities of 2.71 and 3.98 W/cm$^2$ are selected at web speeds of 2.8 and 4 m/min. Subsequently, the drying process is completed within the 12-times-longer drying segment of convection drying. This allows more time for diffusion processes to compensate for the capillary transport of binders and carbon black [12,20]. Additionally, in parameter set V, the second drying chamber is deactivated, reducing the convection drying segment to 1.05 m. This modification aims to investigate a shorter drying segment of the low-drying-rate convection drying. The conceptual representation of the hybrid-drying approach according to the occurring drying stages is illustrated in Figure 3:

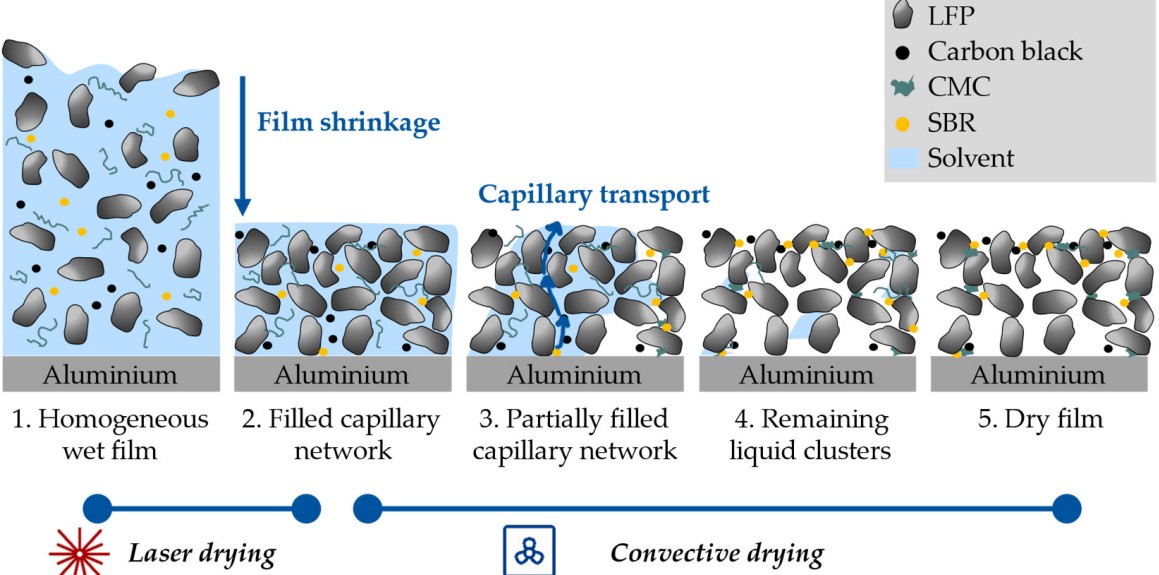

**Figure 3.** Microstructure formation according to JAISER et al. [21] and hybrid-drying concept.

### 2.3. Quality Tests

Adhesion tests are integral to the microstructural analysis of dried electrodes as they indirectly quantify binder migration [11]. In this work, adhesion forces at the contact surfaces between the substrate and the coating are measured using a peel-off test. Cathode samples with a diameter of 46 mm were prepared and analyzed using an adhesion testing device (5944 Single Column Table Top, 5940 Series, Instron GmbH, Darmstadt, Germany). Samples are placed between two orthogonal stamps which are covered with double-sided adhesive tape (Double-sided Universal, Tesa SE). The upper stamp is pressed on the lower stamp with a force of 930 N for 5 s. Then, the two stamps are pulled apart at a speed of 100 mm/min such that delamination between the substrate and the coating occurs. Meanwhile, the maximum tensile stress is measured. In total, five measurements are performed per parameter set and the results are averaged.

To evaluate the electronic conductivity ($\sigma$) of the electrodes, the through-plane resistance ($R$) was measured using a universal testing machine (ZwickiLine Z 2.5 with 200 N Xforce load cell, Zwick/Roell GmbH & Co., KG, Ulm, Germany). Different pressures were applied to the electrode sheets between two copper stamps. For data evaluation, a contact pressure of 30.2 N/cm$^2$ was used according to AMBROCK et al. [22]. The corresponding resistances $R$ were measured with an ohmmeter (Resistomat Type 2316, Burster Präzisionsmesstechnik GmbH & Co. KG, Gernsbach, Germany). Electronic conductivity is calculated

from Equation (1), where *l* is the thickness of the composite electrode and *A* is the contact area of the stamps (6.45 cm$^2$). The method was adapted from Westphal et al. [23].

$$\sigma = \frac{l}{R \cdot A} \qquad (1)$$

For electrochemical characterization in a coin cell setup (CR2032), electrode sheets were additionally dried for 10 h under vacuum at 80 °C to remove water residues, and afterwards punched and handled under dry room conditions. Different laser-/convection-dried LFP-based electrodes were used as positive electrodes, and lithium metal as negative electrodes. A monolayer polypropylene (Celgard 2500, Celgard LLC, Charlotte, NC, USA) was used as separator, and 50 μL 1 M LiPF$_6$ in ethylene carbonate and ethyl methyl carbonate (3:7 by weight, BASF, Ludwigshafen, Germany) were added as electrolyte. Cell assembly was conducted in dry atmosphere with a dew point below −60 °C.

Electrochemical investigations were conducted on a battery tester (Series 4000, Maccor Inc., Tulsa, OK, USA) at 20 °C in a voltage window between 2.5 and 3.8 V. Following a resting step of 4 h, two formation charge/discharge cycles at 0.1 C were implemented. Afterwards, rate capability was investigated using each three cycles of constant current (CC)/constant voltage (CV) charging at 0.2 C with a cut-off current of 0.05 C for CV at 3.8 V and 0.2, 0.5, 1, 2, and 5 C CC discharging.

## 3. Results

### 3.1. Production-Related Results

In the initial stage of the experimental series, the fundamental effectiveness of the hybrid-drying approach was investigated. Therefore, the dried electrodes are monitored inline for major production-side issues such as incomplete drying or visible surface defects such as cracking. The residual moisture of the material is influenced quickly by reabsorption from the ambient air. As a result, all samples measured offline have a residual moisture of between 3 and 5%. Exemplary dried cathodes after the different drying processes are shown in Figure 4.

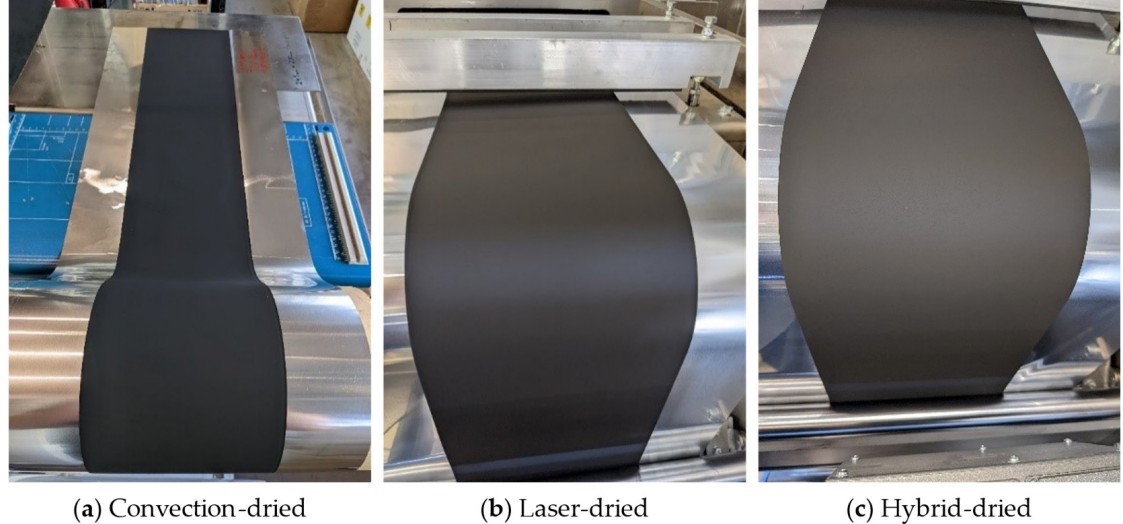

| (**a**) Convection-dried | (**b**) Laser-dried | (**c**) Hybrid-dried |

**Figure 4.** LFP-based cathodes manufactured by the different drying technologies: (**a**) Convection-dried, (**b**) Laser-dried, (**c**) Hybrid-dried.

Within the investigated range of process parameters, no discernible distinctions are observed among the convection-based, laser-based, and hybrid-drying ones. All manufactured electrodes exhibit homogeneous drying and show a similar optical appearance free of surface damages or conspicuous irregularities. Notably, the hybrid-drying process achieves significantly enhanced web speeds of up to 4 m/min, effectively doubling the maximum

achievable process speed of 2 m/min for convection drying. Furthermore, when employing parameter set V, where the second drying chamber is deactivated and the convection drying segment was reduced by half (1 HZ = 1 heating zone), a web speed of 2.8 m/min could be attained. These findings not only underline the fundamental effectiveness of hybrid drying but also highlight its substantial potential to enhance electrode manufacturing productivity with regard to space requirements and production speeds. Figure 5 provides a graphical representation of the achieved production speeds for different drying techniques, alongside the corresponding lengths of the drying segments.

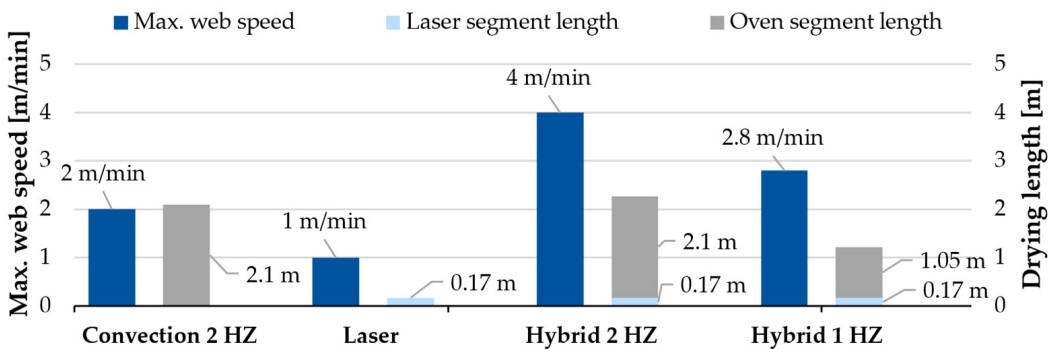

**Figure 5.** Maximum attainable web speed and drying segment lengths of the process alternatives in the given experimental setup.

*3.2. Adhesion*

The adhesive forces of the electrodes produced by the examined process parameters are shown in Figure 6. The adhesion value of 50.2 N/cm$^2$ for the uncalendered electrodes dried by convection (parameter I) serves as a benchmark as it marks a typical value for convection-dried electrodes in the given experimental setup. In contrast, the adhesion of laser-dried electrodes (parameter II) decreased significantly to 26.8 N/cm$^2$. This can be attributed to the significantly higher drying rates of laser-only drying in all drying stages compared to convection drying. While parameter I had a drying time of 70 s, laser-only drying required 12.8 s. Faster drying rates, particularly in the later drying stages, provide less time for diffusion processes to compensate for the capillary transport of the binders [12,20]. As a result, the accumulated binders at the electrode surface decrease the mechanical properties of the dried electrode [11–13].

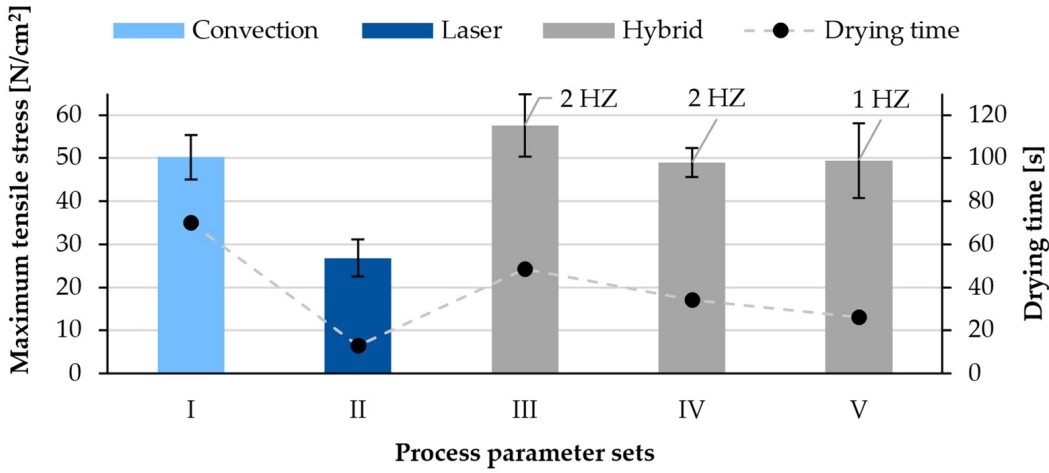

**Figure 6.** Adhesion and drying times of LFP-based cathodes produced under varying drying conditions.

Regarding hybrid-dried electrodes, it can be observed that adhesion was determined within the same range as the benchmark convection-dried electrodes. This applies to all

hybrid-dried electrodes, even though their drying times were reduced by 31% (parameter III), 51% (parameter IV), or 63% when one heating zone was turned off (parameter V) compared to the 70 s drying time of the convection-based benchmark process (parameter I). Thus, it can be demonstrated that employing high-drying-rate laser drying solely during film shrinkage, followed by low-drying-rate convection drying, enables overall increased drying rates without compromising the mechanical properties of the electrode.

### 3.3. Electronic Conductivity

The electronic conductivity $\sigma$ of differently dried LFP-based electrodes are depicted in Figure 7. For laser-dried electrodes, a significantly lower $\sigma$ was determined in comparison to convection-dried electrodes, whereas hybrid-dried electrodes yield similar $\sigma$ compared to the benchmark. Comparing the hybrid-dried electrodes, no significant difference could be observed for $\sigma$.

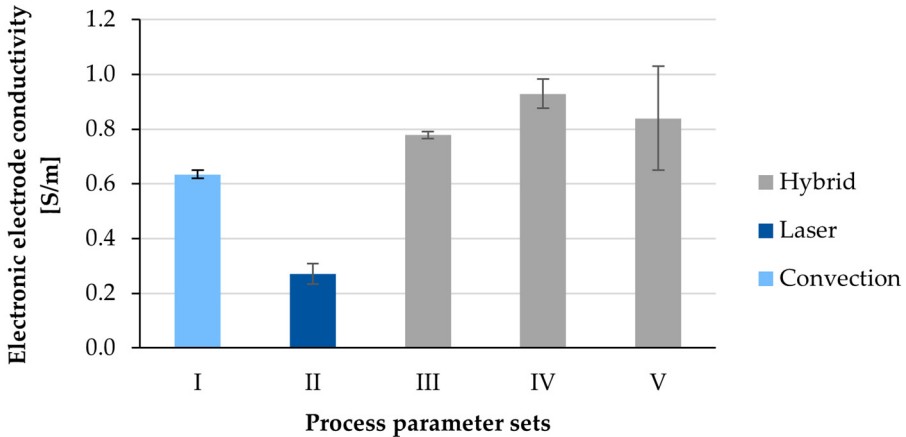

**Figure 7.** Electronic through-plane conductivity of LFP-based positive electrodes produced under varying drying conditions at a contact pressure of 30.2 N/m$^2$.

A lower electronic through-plane conductivity could be attributed to binder and carbon black migration. During fast laser drying, the dissolved binder gets carried with the solvent to the electrode surface, resulting in an uneven distribution of the insulating binder within the composite electrode [24]. Thereby, electronic resistances are increased at the electrode surface, yielding a lower overall electronic conductivity. Beyond that, a reduction of the conductive additive content in electrode regions close to the current collector could further reduce the overall electronic conductivity and, thus, potentially the electrochemical performance.

### 3.4. Electrochemical Performance

To gain information about the impact of varying drying processes and production parameters on electrochemical performance, the rate capability of the electrodes was investigated in an LFP||Li setup and the results are shown in Figure 8. It can be stated that no significant differences were determined for either drying approach. A decreased adhesive strength, as well as the lower electronic conductivity of laser-dried electrodes in comparison to the other manufactured electrodes, does not seem to have an impact on the rate capability in the investigated C-rate range. An uneven binder distribution and the consequential decreased adhesive strength presumably affect long-term cell performance as this results in enhanced capacity fading due to the mechanical deterioration of the composite electrode [20]. Furthermore, decreased electronic conductivity might not have an impact on rate capability due to the overall high amount of carbon black within the electrode (9.3 wt%) that could compensate for inhomogeneities to a considerable degree, as an excessive amount of conductive additive might guarantee sufficient electronic conductivity in composite electrode regions close to the current collector. The results of the rate capability

tests show that the measured electronic conductivities for laser-dried electrodes exceeds a critical value and enough electrons are available for charge-transfer reactions during cycling, as the rate capability is limited to the ionic conductivity of the composite electrode next to its electronic conductivity [25].

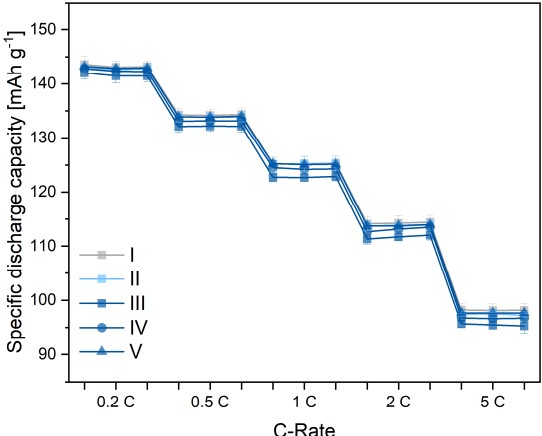

**Figure 8.** Rate capability of the LFP-based positive electrodes produced under varying drying conditions at constant 0.2 C CCCV charging and discharging at varying C-rates for three cycles each.

## 4. Discussion

The comparison of the different drying technologies based on adhesion, electronic conductivity, and rate capability demonstrated the high suitability of hybrid-drying technologies by laser and convection. As supported by JAISER et al., the adhesion of the electrode coating was found to be highly dependent on the drying parameters [14]. The results also confirmed that binder migration occurs primarily due to capillary forces in the drying phases following film shrinkage. However, pre-drying by laser with its high energy input at shorter drying times, followed by convection-based drying, did not affect the electrode adhesion.

Electronic through-plane conductivity measurement results suggested enhanced carbon black and binder migration for exclusively laser-dried electrodes; however, hybrid-dried electrodes did not exhibit such detrimental effects, proving that laser drying can be successfully implemented in the evaporation zone of the drying process.

Rate capability tests showed no significant difference between all five drying process parameter sets, which implies that the proposed laser-drying technique might be applicable for high-power LFP-based electrodes with comparatively low areal capacities. In future studies, the long-term cycling behavior of hybrid- and laser-dried electrodes should be investigated. This will help uncover the potential impact of the inhomogeneous binder and carbon black distribution within laser-dried composite electrodes on electrochemical performance and capacity fading in comparison to the benchmark, as well as hybrid-dried electrodes.

In addition, other material compositions (e.g., NMC and LCO) and mixing protocols could influence the electrode quality. Mixing phases with high shear rates, for instance, can lead to the decomposition of the SBR. This can be prevented by adding the SBR in later mixing phases.

Future research should also explore the applicability of the drying techniques for higher areal loadings respectively film thicknesses to assess the suitability of laser- and convection-based hybrid drying for high-energy electrodes. Moreover, the utilization of waste heat from the extracted moist air in an energy recovery system in the convection dryer could further optimize the energy efficiency of the drying process. This could be extended by implementing a closed-loop inline-controlled laser-drying system to compensate for inhomogeneities in the drying process and reduce defects during ramp-up after machine downtimes. This could be supported by the use of machine-learning algorithms in combi-

nation with the implementation of appropriate inline sensor technology. The evaluation of the energy-saving potential and the impact on CapEx and OpEx in the drying process compared to other drying technologies such as IR, convection, or induction should be part of future studies. Lastly, an investigation of the transferability of the hybrid-drying results from the laboratory to the industrial scale, along with process-side implications for future battery production, is recommended to determine the practical viability of laser-drying technology with an optimized intensity profile in industrial applications.

**5. Conclusions**

This study performed with aqueously processed LFP-based cathodes demonstrates the high potential of a hybrid-drying approach using laser- and convection-based processes in battery production. The hybrid process significantly decreases manufacturing costs through faster drying rates compared to conventional convection-based drying while maintaining comparable electrode and cell quality. To evaluate electrode and cell quality, the adhesion and electronic conductivity of hybrid-dried LFP-based cathodes were compared with reference cathodes dried solely through convection or exclusively by high-power diode lasers. The detrimental adhesion loss caused by binder migration in purely high-power laser-dried electrodes can be mitigated by employing the hybrid-drying process. This study reveals that utilizing laser-based drying during the initial drying phases of film shrinkage results in similar adhesion, electronic conductivity, and rate capability outcomes compared to convection-based drying. Additionally, the hybrid-drying process reduces drying times by up to 63% compared to state-of-the-art convection-based drying, enabling substantial throughput enhancements in industrial-scale production with the simultaneous reduction of the system footprint.

For further process optimization and compensation for potential drying inhomogeneities, future research should focus on inline control of the process. A closed-loop control concept could be implemented to optimize the drying process in terms of energy efficiency and drying quality. In addition, battery cells with hybrid-dried electrodes by laser and convection should be subjected to long-term cycle tests to assess potential capacity losses throughout their cycle life.

**Author Contributions:** Conceptualization, S.W., N.S., T.T. and V.G.; methodology, S.W., N.S., T.T. and V.G.; investigation, N.S., T.T., S.W. and V.G.; writing—original draft preparation, S.W., N.S., T.T. and V.G.; writing—review and editing, S.W., V.G. and M.B.; visualization, S.W., N.S., T.T. and V.G.; supervision, D.N., M.B., H.H., M.W. and A.K.; project administration, S.W., M.B., M.W. and A.K.; funding acquisition, D.N., H.H., M.W. and A.K. All authors have read and agreed to the published version of the manuscript.

**Funding:** This work is part of the Project IDEEL. This research was funded by the Federal Ministry of Education and Research, grant number 03XP0414.

**Data Availability Statement:** Not applicable.

**Acknowledgments:** The authors would like to thank the other project partners Fraunhofer FFB, Fraunhofer ILT, Laserline GmbH, Optris GmbH, and Coatema GmbH for their co-operation and expert advice in the project IDEEL.

**Conflicts of Interest:** The authors declare no conflict of interest.

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
