# Peer review of "Optimized LiFePO4-Based Cathode Production for Lithium-Ion Batteries through Laser- and Convection-Based Hybrid Drying Process"

_wevj, doi:10.3390/wevj14100281_

Round 1

Reviewer 1 Report

This paper demonstrates an improved method of hybrid LFP cathode drying process, which utilizes laser and convection to improve efficiency and operational costs. The authors compared the proposed method with laser & convection method in terms of adhesion, electronic conductivity, and rate capability, with a clear description of the methodology & characterization. In addition to the proposed method, the author could consider adding a potential up-scaling approach of the proposed method for large-scale applications, and demonstrate if there are less risk of up-scaling compared to conventional laser & convection method, which would further validate the advantages of the proposed hybrid method.

Reviewer 2 Report

In this manuscript, laser drying process is hybrid with convection drying process in making LFP cathode electrodes. Different parameters are tested. The results showed that hybrid laser drying with convection drying can reduce drying time and improve through plan conductivity as well as adhesion. However, some questions need to be addressed or answered before it can be published.

1.       Did author compare the moisture level after different drying conditions? Only the images in Figure 4 is not enough to show the electrodes are fully dried.

2.       Is there any thickness or porosity difference between different drying condition?

3.       Did author do electrode calendar before assembling cells?

4.       Why the electrical conductivity shows bid difference while rate performance is same? Any evidence shows that “high amount of carbon black within the electrode compensates inhomogeneities?

5.       It would be better show long cycle stability difference. Otherwise, the data is not complete and the difference in electric conductivity and peal test didn`t result in electrochemical performance difference is not make sense.

6.       Does the convection heating zone come with IR heater?

7.       In experiment part, author add CMC dry powder directly in slurry mixing. Why not pre-mix CMC solution? CMC is hard to dissolve in water, usually 2% takes several hours in heating condition. How can author make sure CMC is well dispersed?

8.       In table 1, can author double check the number for SBR? Seems all the other component is change simultaneously comparing wt% in solid and total, but SBR didn`t change too much.

Reviewer 3 Report

Summary:

This Research Article wevj-2624018 titled, “Optimized LiFePO4-based Cathode Production for Lithium-Ion Batteries through Laser and Convection-based Hybrid Drying Process,” reports on the investigation and subsequent optimization of electrode drying processes for lithium-ion batteries. The effect of laser-based drying processes is investigated and compared with the conventional heating method on the cell properties. Accordingly, hybrid systems combining laser- and convection-based drying are investigated in an experimental study with water-processed LiFePP4 cathode.

General comment:

The research reports the study and design of the fabrication method of lithium-ion battery cathode as well as the corresponding material, electrochemical, and cell performance. Thus, I would like to suggest the acceptance of this research, and some minor revisions are suggested to provide the necessary data and explanation. I hope the authors find the comment useful.

Comments:

(1) In general, LiFePO4 is not a promising cathode for the development of the battery technology. Would the proposed method to be applied to the LCO or NMC cathode? What would happen if the proposed method is adopted for the graphite anode application?

[Suggestion] The proposed method is interesting. It is suggested to make an assumption if the method is applied to electrode with good performance, while having challenges in fabrication.

(2) In the analysis and discussion, some electrode properties are reported in the manuscript. However, it is suggested to report the solid experimental data with the support of SEM and AFM analysis to show the morphology and surface of the electrode. EDS and the related mapping analysis are suggested to prove the uniform distribution of elements in the electrode.

[Suggestion] Please consider providing material analysis of the cathode to support the discussion.

(3) In the analysis and discussion, the electrochemical and cell analyses are important data and should be reported in this work. First, the rate capability should have a longer testing cycle number than 3 cycles in each rate. Moreover, the discharge/charge efficiency is needed to be reported. Second, cyclability is necessary for LFP cathode. Third, EIS and CV data as well as the discharge/charge voltage profiles are necessary to make the comparison of the effect of the fabrication methods toward the cathode performance.

[Suggestion] Please consider providing the necessary electrochemical and cell analysis.

(4) In the reference section, it is found that the cited reference papers are highly similar to the corresponding author’s previous paper (Process and Material Analysis of Laser- and Convection-Dried Silicon–Graphite Anodes for Lithium-Ion Batteries). Moreover, some similar content and writing in this submission and the author’s previous manuscript should be rewritten.

[Suggestion] Please update your reference and rewrite the sentences that are similar to the published work.
